# Occupational exposures among women *beedi* workers in Mysore District, India: A mixed-methods study protocol

**Priyanka Ravi**[1]*, **Kiranmayee Muralidhar**[2,3], **Purnima Madhivanan**[1,2], **Amanda M. Wilson**[4], **Frank A. von Hippel**[4], **Amina Salamova**[5], **Eva Moya**[6], **Lynn B. Gerald**[7]

1 Department of Health Promotion Sciences, Mel & Enid Zuckerman College of Public Health, University of Arizona, Tucson, Arizona, United States of America, 2 Public Health Research Institute of India, Mysore, India, 3 JSS Academy of Higher Education and Research, Mysore, India, 4 Department of Community, Environment & Policy, Mel & Enid Zuckerman College of Public Health, University of Arizona, Tucson, Arizona, United States of America, 5 Rollins School of Public Health, Emory University, Atlanta, Georgia, United States of America, 6 Border Biomedical Research Center, The University of Texas, El Paso, Texas, United States of America, 7 Population Health Sciences Program, University of Illinois, Chicago, Illinois, United States of America

* priyankaravi@arizona.edu

**Data Availability Statement:** No datasets were generated or analysed during the current study. All

## Abstract

*Beedi* is the most common smoking form of tobacco used in India. The rolling of *beedis* is performed primarily by women in settings that lack occupational safeguards. The aims of this protocol are to establish methods for the study of occupational exposures among women *beedi* workers and their experiences and challenges working with unburnt tobacco. This protocol employs a convergent parallel mixed-methods approach. Qualitatively, we plan to explore the experiences and challenges faced by women *beedi* workers using photo-voice, a community based participatory method. Occupational exposures to pesticides will be assessed through the use of silicone wristbands worn for seven days by workers, and exposure to toxic metals and metalloids will be assessed in dust samples collected in the homes of workers. The outcomes will be analyzed to form policy recommendations to improve the occupational health of women beedi workers.

## Introduction

A hand-rolled tobacco cigarette in India and Bangladesh is called a *beedi*, whereas in Indonesia it is referred to as *kretek* [1–3]. The process of making *beedis* includes collecting *tendu* leaves to serve as the wrapper and then rolling the tobacco within them. The *beedis* are then sorted, labelled, and packaged. Most of the *beedi* workers are women [4]. In India, the number of *beedi* workers employed in the registered sector varied from 0.4 million in 2000–2001 and 2005–2006 to 0.3 million in 2010–2011 [5]. The unregistered sector numbers varied from 3.1 million workers in 2000–2001 to 4.1 million workers in 2005–2006 and 2.9 million workers in 2010–2011 [5]. In Bangladesh, the Bureau of Statistics estimated the number of *beedi* workers at 266,818, while the tobacco industry claimed to employ 2.5 million workers [3]. In Indonesia,

relevant data from this study will be made available upon study completion.

**Funding:** NIOSH ERC Pilot Project Research Training (PPRT) Program Grant, UCLA and Deans Dissertation Grant, University of Arizona.

**Competing interests:** The authors have declared that no competing interests exist.

the *kretek* manufacturing industry employed over 260,000 workers in 2006 [2]. In India, men are primarily employed in the factory system of the tobacco industry, whereas 90% of women employees work in the home-based system of *beedi* making [6]. The working conditions in home-based *beedi* rolling typically include conditions such as poor lighting, poor ventilation, and overcrowding [7]. *Beedi* workers earn considerably less income compared to workers in other manufacturing industries, further subjecting them to income inequality [5].

Many types of pesticides are used in tobacco farming, including some containing heavy metals and others containing toxic compounds such as organophosphates; workers are exposed to these during the *beedi* rolling process [8]. Pesticides used in tobacco fields are associated with numerous adverse health outcomes. For example, epigenetic alterations were reported in tobacco farmers occupationally exposed to mixtures of pesticides [9]. *Beedi* rollers face the occupational hazards of exposure to both pesticides and tobacco [10].

Occupational exposure to tobacco dust among women *beedi* workers in Mangalore, India was associated with risk of developing cervical cancer [11]. Women *beedi* workers in Telangana, India were found to have low literacy and poor awareness of occupational health hazards and hygienic practices [4]. Workers engaged in tobacco cultivation often suffer from an occupational illness known as "green tobacco sickness" (GTS), found to be caused by the absorption of nicotine from wet tobacco plants [12]. *Beedi* rolling during pregnancy reduces cord blood leptin levels independent of birthweight and induces reduced size for gestational age [13]. Low birth weight babies, treatment for infertility, and premature menopause were also reported by women working in *beedi* industry [7]. Women *beedi* workers in Patna, India were found to have low haemoglobin, neutrophils, and monocytes, and increased lymphocytes and eosinophils as compared to non-*beedi* workers [14].

Silicone wristbands serve as sensitive passive samplers to environmental exposures to pesticides [15, 16]. Some pesticides, including organophosphates and pyrethroids, are present in house dust near areas of use [17, 18]. Vacuum sampling is an effective means to collect residential dust samples to detect pesticides and toxic metals and metalloids [19].

Published studies on the occupational health of *beedi* workers are quantitative in nature and have not considered the lived experiences of women working in this industry that can be assessed using qualitative research. Most past research has not adequately involved participants and considered occupational challenges from a theoretical perspective. Mixed methods approaches are needed to incorporate exposure data with information on lived experiences of women *beedi* workers in order to improve working conditions and inform effective policy. The purposes of this protocol are to establish methods for the study of occupational exposures among women *beedi* workers and their experiences and challenges working with unburnt tobacco.

Aim 1: Explore the experiences and challenges faced by women *beedi* workers in Mysore, India using photovoice, a community based participatory method. Twenty participants will be recruited for the photovoice discussion.

Aim 2: Determine the occupational exposures among women *beedi* workers in Mysore, India. Participants will include 30 women *beedi* works and 30 age and socioeconomic status matched non-*beedi* workers. Occupational exposures will be estimated via analysis of a silicone wristband worn for seven days during working hours. In addition, dust samples will be collected at the study participants' homes to estimate pesticide exposure in the households.

## Methodology

A complex mixed method participatory design [20] with community-based participatory research (CBPR) methods such as photovoice will provide the necessary framework for the

proposed study. CBPR incorporates culturally relevant research models that address issues of importance to the community. Photovoice is a CBPR method that can be used to foster trust and capacity building for community led solutions to environmental and health related issues [21]. As an emerging CBPR methodology, photovoice promotes social action by equipping communities to participate in the identification and analysis of local problems [22]. The photovoice method is premised on three core goals of the research process [23]: (1) enable people to record and reflect on their community's strengths and concerns, (2) promote critical dialogue and knowledge about important issues through large and small group discussions of photographs, and (3) reach policy makers. In addition to employing photovoice, we will use silicone wristbands and collect dust samples from homes to examine pesticide and toxic metal exposures, respectively, among women working in *beedi* rolling industry.

A convergent parallel core design will be used, which is a type of study in which qualitative and quantitative data are collected in parallel, analyzed separately, and then merged [20]. The quantitative data will be used to assess exposures to pesticides and toxic metals derived from tobacco dust processed in the homes of participants. The qualitative data will explore the experiences and challenges faced by the women *beedi* workers handling unburnt tobacco. The combined datasets will enhance insights on occupational health concerns of women *beedi* workers and facilitate the development of improved policies.

## Approach

A mixed methods parallel core design [20] (Fig 1) will be used to determine the occupational exposures among women *beedi* workers, and their experiences and challenges working with unburnt tobacco. The qualitative and quantitative data will be collected in parallel, analyzed separately, and then merged. Mysore District, Karnataka, India was chosen as the study site because Mysore has a large *beedi* worker community [24, 25]. The research will be facilitated by the Public Health Research Institute of India (PHRII), a non-profit organization in Mysore.

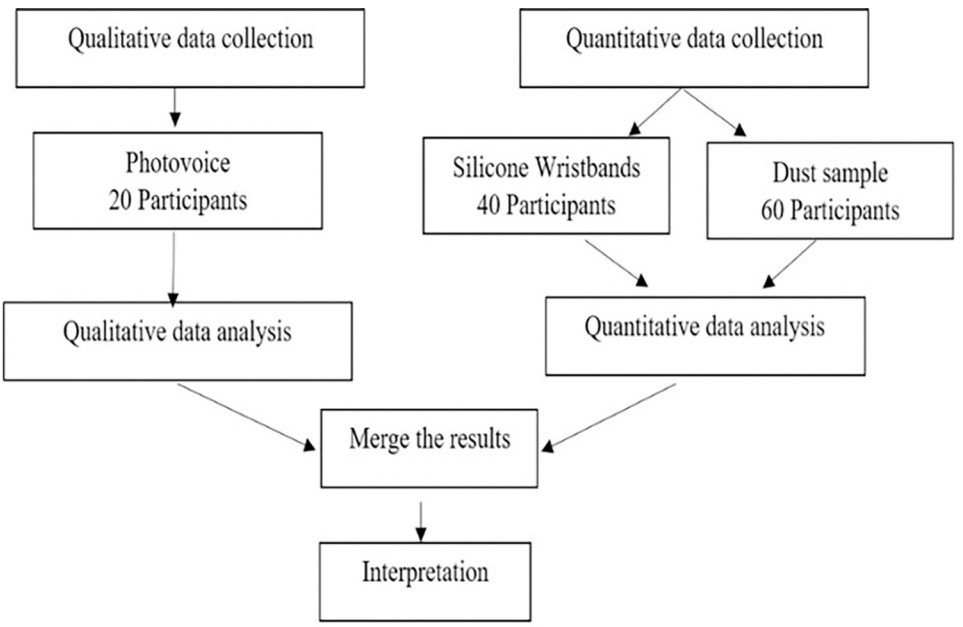

**Fig 1. Mixed methods parallel core design.**

PHRII has provided access to health services for women in Mysore since 2010. PHRII will assist in participant recruitment as they are known to the population of interest.

**Qualitative phase.** Participants will be recruited from the *beedi* worker community in Mysore District. A maximum of 20 study participants will be enrolled in the study for photovoice. Two photovoice discussions will be conducted with ten individuals in each group. Inclusion criteria will be women *beedi* workers who are 18 years and above (one woman per household), the ability to write in either Kannada or English, the ability to text and take photos on a mobile phone or digital camera, willingness to undergo the informed consent process, and willingness to be audio-recorded during the photovoice discussion sessions. Exclusion criteria will be inability to speak Kannada or English, currently pregnant, and unwillingness to consent to participate in the study.

A representative from PHRII will first consult with community leaders to obtain permission to carry out the project in their community. Then a PHRII staff member will reach out to the community health workers (CHW) in these communities where we propose to recruit participants. The CHW living in these communities are well connected to the residents and therefore their assistance will facilitate recruitment. A flyer about the study will also be distributed through the CHW during their regular house visits. Furthermore, PHRII has conducted several studies in these communities, and has established relationships with residents. If any potential participant expresses interest to the CHW, she will be referred to the PHRII study recruiter. The study recruiter will then assess eligibility and if the woman is eligible, she will be asked to undergo the informed consent process. The informed consent form will be read aloud to the potential participants in Kannada, and the women will be encouraged to ask questions or express any doubts about the study before they are invited to sign the informed consent form.

The study will be conducted at the PHRII conference room. The photovoice discussions will be conducted outside of regular working hours, when it is convenient for the participants to actively engage and participate in the process. The photovoice participants will be reimbursed in local currency at the equivalent of USD$21 per person for their time and effort in all the photovoice sessions.

Information will be collected on demographic details, occupational history, and occupational safety practices including handwashing and wearing gloves and masks. The photovoice will be conducted over a period of four weeks (Table 1). In the first week, a one-day training on skills needed to implement photovoice will be provided for the 20 participating *beedi* workers. Training topics will include introduction to photovoice, principles and ethics of photography, photo consent and safety, field practice using digital cameras, and group discussion and reflection about the photovoice process and outcomes. The first component of photovoice will involve reaching consensus on the research topic with the collaborating community. This initial phase is a crucial first step to facilitate meaningful partnerships and engagement and counter power imbalances within the team [21]. The topics for photovoice will be decided after discussing with the participants during the training session. Participants will then be asked to take pictures of what for them best represents occupational health challenges in their workplace. Photos will be taken over seven days after which the participants will transfer the photos to the research team.

In weeks two, three, and four participants will first share the pictures with the research team, participants will first complete a SHOWeD [26] form and then discuss their two chosen photographs to elicit in-depth discussion and narration. SHOWeD form includes photo discussion prompts questions such as: 'What do you see?', 'What is really happening?', 'How does this relate to our life?', 'Why does this situation, concern, or strength exist?', What can we do to educate others about this situation, concern, or strength?', and What can we do about it'

Table 1. Overview of photovoice timeline and activities.

| Week | Activity |
|------|----------|
| Week 1 | Photovoice Training<br>• Introduction to photovoice and project goals<br>• Photographic techniques<br>• Confidentiality and ethics |
| Week 2 | Photo collection<br>• Each participant will collect five occupational health related theme photos and share them with research staff<br>Discussion 1<br>• SHOWeD form completion<br>• Participant presentations<br>• Group discussion |
| Week 3 | Photo collection<br>• Each participant will collect five occupational health related theme photos and share them with research staff<br>Discussion 1<br>• SHOWeD form completion<br>• Participant presentations<br>• Group discussion |
| Week 4 | Photo collection<br>• Each participant will collect five occupational health related theme photos and share them with research staff<br>Discussion 1<br>• SHOWeD form completion<br>• Participant presentations<br>• Group discussion<br>Wrap-up and celebration<br>• Comprehensive slideshow with all photos and final narratives will be shared with the group<br>• Evaluation survey |

[26]. Discussion sessions will begin with individual presentations of the photographs. Participants will be asked to explain the background of the picture, how it relates to the topic, and what it means for them. The facilitator will promote the group discussion by prompting: 1) strengths and weaknesses; 2) similarities and differences; and 3) identified and connected themes. During the final week, participants will be asked to vote on the pictures to be selected for the photo exhibition, and participants will be asked to complete a survey to reflect on the most and least preferred aspects of the project in Kannada language (Table 1).

Participants and facilitators will maintain contact between meetings via a WhatsApp group message thread or phone calls. In the WhatsApp group, facilitators will remind participants about meetings and check in mid-week to ask about progress with photo collection. Participants will be encouraged to actively engage in the group chat, and share health challenges at their workplace.

Later, a photovoice exhibition will be arranged within the communities. The community leaders, community members, and policy makers will be invited to view the photos. This will help to raise awareness of the challenges faced by *beedi* workers.

**Data analysis.** An inductive approach will be used to code the data, which will be analyzed using Atlas.Ti 22 software [27]. We will use a constructivist grounded theory approach [28] and SHOWeD [23] checklist for the data analysis [29]. The codes will be generated and modified to create the themes. The researchers will maintain neutrality to ensure the situations are not intentionally influenced or manipulated.

**Quantitative phase.** This is a pilot study to measure the pesticide and toxic metal exposures among *beedi* workers as compared with non-*beedi* workers. The study participants will include 20 women *beedi* workers involved in the qualitative study, an additional 10 beedi

workers not involved in photovoice, and 30 non-beedi workers from the same community involved in a different occupation other than tobacco farming or *beedi* rolling. The participants will be matched by age and socioeconomic status. Inclusion criteria for *beedi* workers will be participants who are current *beedi* workers 18 years or older, willing to wear a silicone wristband for seven days, and allowing the collection of dust samples from their homes. Inclusion criteria for non-*beedi* workers from the same community will be women involved in any other home-based work (including unemployed), not previously involved in *beedi* rolling, willing to wear a silicone wristband for seven days, and allowing the collection of dust samples from their homes. Exclusion criteria for beedi and non-beedi workers will be women who are currently pregnant or not willing to provide written informed consent. All the participants will be reimbursed in local currency the equivalent of USD$4 for their time and effort.

**Silicone wristbands.** We will use forty low-cost silicone wristbands as non-invasive passive samplers to assess cumulative seven-day exposures to pesticide residues among randomly selected *beedi* and non-*beedi* workers (n = 20 in each group). Participants will be provided with a pre-cleaned silicone wristband wrapped in foil and sealed in a Ziplock bag [30]. Participants will wear the silicone wristbands on their dominant hand continuously for seven days. After seven days, PHRII team members will collect the wristbands, wrap them in foil, seal in a Ziplock bag, and store them in a -20˚C freezer. The silicone wristbands will be shipped to Emory University for analysis of organochlorine pesticides.

**Dust samples.** We will collect dust samples from the homes of all 60 participants. All the samples will be collected from entirety of the living room and bedroom and the dimensions of the room will be measured. The dust sample will be collected over a 40 minute sampling period in each home with a vacuum outfitted with dust collection paper bags. The bags containing dust samples will be wrapped in foil and stored in a Ziplock bag at -20˚C. Each dust sample will be labelled with sampling location and date. The dust samples will be shipped to the University of Arizona for quantification of toxic metals.

**Data analysis.** We will estimate the exposure to organochlorine pesticides from the silicone wristbands and to toxic metals from the dust samples. We will compare exposures between *beedi* and non-*beedi* rolling participants using a variety of parametric and nonparametric approaches, depending upon characteristics of the data. The quantitative and qualitative study findings will be compared in order to inform effective policies to protect the health of women *beedi* workers. At the end of the study, we will go back to the community and share the study results with the participants and community leaders.

## Limitations

The non-*beedi* workers (control group) could experience different environmental exposures to pesticides and toxic metals, which could bias our study findings. We will attempt to minimize bias by matching age and socioeconomic status. The participants will be asked to wear the silicone wristbands for seven continuous days. We will try to minimize non-compliance by checking in with participants during the seven-day period. We will collect dust samples at one point in time so if the participant had cleaned the house recently then the dust would represent an underestimation of metal exposures in the home; therefore, we will ask participants not to clean the house one day prior to the dust sample collection.

## Supporting information

**S1 Checklist. *PLOS ONE* clinical studies checklist.**
(DOCX)

## Author Contributions

**Conceptualization:** Priyanka Ravi, Kiranmayee Muralidhar, Purnima Madhivanan, Frank A. von Hippel, Amina Salamova, Eva Moya, Lynn B. Gerald.

**Data curation:** Priyanka Ravi.

**Formal analysis:** Priyanka Ravi, Purnima Madhivanan, Amanda M. Wilson, Frank A. von Hippel, Eva Moya.

**Funding acquisition:** Priyanka Ravi, Amanda M. Wilson, Lynn B. Gerald.

**Investigation:** Priyanka Ravi, Kiranmayee Muralidhar, Purnima Madhivanan, Frank A. von Hippel.

**Methodology:** Priyanka Ravi, Kiranmayee Muralidhar, Purnima Madhivanan, Amanda M. Wilson, Amina Salamova, Eva Moya, Lynn B. Gerald.

**Project administration:** Kiranmayee Muralidhar.

**Resources:** Frank A. von Hippel.

**Supervision:** Purnima Madhivanan, Frank A. von Hippel.

**Validation:** Amina Salamova, Eva Moya.

**Visualization:** Eva Moya.

**Writing – original draft:** Priyanka Ravi, Kiranmayee Muralidhar.

**Writing – review & editing:** Priyanka Ravi, Purnima Madhivanan, Amanda M. Wilson, Frank A. von Hippel, Amina Salamova, Eva Moya, Lynn B. Gerald.

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
