## [Decision Letter · Decision Letter 0]

2 Nov 2023

PONE-D-23-17775Occupational exposures among women beedi workers in Mysore District, India: A mixed-methods study ProtocolPLOS ONE

Dear Dr. Ravi,

Thank you for submitting your manuscript to PLOS ONE. After careful consideration, we feel that it has merit but does not fully meet PLOS ONE’s publication criteria as it currently stands. Therefore, we invite you to submit a revised version of the manuscript that addresses the points raised during the review process.

We look forward to receiving your revised manuscript.

Kind regards,

Adekunle Akeem Bakare, Ph.D.

Academic Editor

PLOS ONE

Journal Requirements:

Additional Editor Comments :

Please note that as this is a Study Protocol, it is not necessary to include Results.

Reviewers' comments:

Reviewer's Responses to Questions

**Comments to the Author**

1. Does the manuscript provide a valid rationale for the proposed study, with clearly identified and justified research questions?

Reviewer #1: Yes

Reviewer #2: Yes

Reviewer #3: Partly

2. Is the protocol technically sound and planned in a manner that will lead to a meaningful outcome and allow testing the stated hypotheses?

Reviewer #1: Yes

Reviewer #2: Yes

Reviewer #3: Partly

3. Is the methodology feasible and described in sufficient detail to allow the work to be replicable?

Reviewer #1: Yes

Reviewer #2: Yes

Reviewer #3: Yes

4. Have the authors described where all data underlying the findings will be made available when the study is complete?

Reviewer #1: No

Reviewer #2: Yes

Reviewer #3: Yes

5. Is the manuscript presented in an intelligible fashion and written in standard English?

Reviewer #1: Yes

Reviewer #2: Yes

Reviewer #3: Yes

6. Review Comments to the Author

You may also provide optional suggestions and comments to authors that they might find helpful in planning their study.

Reviewer #1: a) Line 71&73: as the study is proposed in a subnational unit of India; Bangladesh, and Indonesia, data seems unconnected in this context. In case you still feel it is needed, mention why these two countries are taken as references.

b) Line 90: ‘Occupational exposure to tobacco dust among women beedi workers in Mangalore, Karnataka’.

Kindly mention the country; international readers may be unable to locate the place. The same applies to Telangana (92) and Patna, Bihar (100). Alternatively, you can also mention all studies together saying, ‘studies conducted in India…’ and then mention all three studies one by one in the existing format.

c) 127: instead of explaining the goals of the photovoice method, it is better to explain the photovoice method and how the study is conducted using it. Although you have mentioned it under ‘Approach’ in detail, it is unclear for a first-time reader what exactly photovoice is.

d) You have not stated why Mysore is selected for the study.

All the best.

Reviewer #2: 1.The sample size is relatively small, and it might be better to have the same sample size among all groups.

2. Beedi workers working duration may differ for each person. It is preferable to select participants with similar work experience.

3. Mentioning the participants' marital status for the specific age group could be more suitable. If married, please mention whether pregnant ladies are included or not in the study group.

4. You could provide more details about the control group and include them in the methodology.

5. In inclusion criteria participants age from 18 to particular age will be more suitable for the study

Reviewer #3: The manuscript is strictly on protocol and essentially on the use of photovoice. Protocol cannot be published in isolation unless the outcomes of the study are included. Much as the protocol will achieve the set objectives, the results of the protocol should be included. The references should also be current.

7. PLOS authors have the option to publish the peer review history of their article (what does this mean?). If published, this will include your full peer review and any attached files.

Reviewer #1: No

Reviewer #2: **Yes: **Yamini Kanipakam

Reviewer #3: No

---

## [Author Response · Author response to Decision Letter 0]

2 Jan 2024

14th December 2023

Tucson, USA

Dear Dr. Adekunle Akeem Bakare, 

Thank you for your thoughtful review of our manuscript PONE-D-23-17775, and for the helpful comments of the three reviewers. I sincerely appreciate and thank you all for taking your valuable time to help us to substantially improve the paper. We have made substantial edits to the entire manuscript, as shown in the revised version with changes highlighted. Below please find our responses to the reviewers' specific comments.

Reviewer #1

Comments Reply

a) Line 71&73: as the study is proposed in a subnational unit of India; Bangladesh, and Indonesia, data seems unconnected in this context. In case you still feel it is needed, mention why these two countries are taken as references. 

We referred to rolled tobacco workers in Bangladesh and Indonesia because we would like to highlight that occupational tobacco exposure is not only a problem in India. We revised the text for clarity.

b) Line 90: ‘Occupational exposure to tobacco dust among women beedi workers in Mangalore, Karnataka’.

Kindly mention the country; international readers may be unable to locate the place. The same applies to Telangana (92) and Patna, Bihar (100). Alternatively, you can also mention all studies together saying, ‘studies conducted in India…’ and then mention all three studies one by one in the existing format. 

Thank you for your comments. We have made these recommended changes.

c) 127: instead of explaining the goals of the photovoice method, it is better to explain the photovoice method and how the study is conducted using it. Although you have mentioned it under ‘Approach’ in detail, it is unclear for a first-time reader what exactly photovoice is. 

We are introducing the term photovoice in the beginning of the methodology section, so we felt it is important for the readers to know what photovoice tries to achieve. We provided more details on the methods of photovoice, including a table of the week-by-week methods, and clarified the language.

d) You have not stated why Mysore is selected for the study. We have now included the reason for selecting the study site.

Reviewer #2

Comments Reply

1.The sample size is relatively small, and it might be better to have the same sample size among all groups. 

The sample size is small because this is a pilot study and the results from this study will help us to plan future studies with large sample sizes. The sample size is different for the participants who will wear silicone wristband (n=40) vs. dust sample collection (n=60) because 

1. Previous studies on passive air sampling using silicone wristbands (Donald et. al., 2016) and indoor dust sampling (Tashakor et. al., 2022) have mostly used 40 to 60 as the sample size.

2. We are getting the silicone wristbands from our collaborator at Emory University and due to the limited funding available we are only able to analyze 40 wristbands.

Donald CE, Scott RP, Blaustein KL, Halbleib ML, Sarr M, Jepson PC, Anderson KA. Silicone wristbands detect individuals' pesticide exposures in West Africa. R Soc Open Sci. 2016 Aug 17;3(8):160433. doi: 10.1098/rsos.160433. PMID: 27853621; PMCID: PMC5108971.

Tashakor M, Behrooz RD, Asvad SR, Kaskaoutis DG. Tracing of Heavy Metals Embedded in Indoor Dust Particles from the Industrial City of Asaluyeh, South of Iran. Int J Environ Res Public Health. 2022 Jun 28;19(13):7905. doi: 10.3390/ijerph19137905. PMID: 35805563; PMCID: PMC9265302.

2. Beedi workers working duration may differ for each person. It is preferable to select participants with similar work experience. We plan to capture the different work experiences, for example those with more than 10 years beedi rolling experience might have different occupational health concerns compared to those rolling beedi for only 2 years.

3. Mentioning the participants' marital status for the specific age group could be more suitable. If married, please mention whether pregnant ladies are included or not in the study group. 

Thank you for the suggestion. The marital status of the participants is recorded in the demographics, but this is not part of our inclusion criteria. We will exclude participants who are currently pregnant because they are a vulnerable population. We have clarified this in the text.

4. You could provide more details about the control group and include them in the methodology. The control group are matched by age and socioeconomic status. Inclusion criteria for non-beedi workers will be women involved in any home-based work, housewife, never involved in beedi rolling, who agree to wear a silicone wristband for seven days and allow the collection of dust samples from their homes (and from the same community). Exclusion criteria for beedi and non-beedi workers will be participants who are currently pregnant or not willing to give written informed consent. We have clarified the text on this section.

5. In inclusion criteria participants age from 18 to particular age will be more suitable for the study We do not have an upper limit for the age because we would like to capture the data on the occupational experience of different ages. This will provide us an opportunity to understand the recent and past health concerns among beedi rolling women.

Reviewer #3

Comments Reply

The manuscript is strictly on protocol and essentially on the use of photovoice. Protocol cannot be published in isolation unless the outcomes of the study are included. Much as the protocol will achieve the set objectives, the results of the protocol should be included. The references should also be current. Thank you for your comments. 

1. This manuscript is submitted under the protocol publication category of PLOS ONE. We will include the results once the study is completed and cite this manuscript as a published protocol. The goal of publishing this protocol is to establish study guidelines prior to data collection as part of the movement in science to ensure research transparency (guidelines that encompass "a range of open practices including registering studies, sharing study data, and publicly reporting research findings"; this will ensure that we remain in line with the proposed methodology.

2. We have included references such as Rustagi et. al., 2001, Yasmin et. al., 2010, and Khanna et. al., 2014 because they justify the need for this study in beedi workers. 

3. We have added new beedi workers studies conducted in Myore, India: Bhat et. al., 2018 and Aliya et. al., 2018.

4. Wang & Burris, 1997 and Wang 2007 are included because they provide the concept and methodology for photovoice.

Once again, we thank you and the reviewers for your thoughtful review process and for helping us to significantly improve the manuscript.

 Sincerely,

 Priyanka Ravi

---

## [Editor Report · Decision Letter 1]

10 Jan 2024

Occupational exposures among women beedi workers in Mysore District, India: A mixed-methods study protocol

PONE-D-23-17775R1

Dear Dr. Ravi,

We’re pleased to inform you that your manuscript has been judged scientifically suitable for publication and will be formally accepted for publication once it meets all outstanding technical requirements.

Kind regards,

Adekunle Akeem Bakare, Ph.D.

Academic Editor

PLOS ONE
---

## [Editor Report · Acceptance letter]

25 Mar 2024

PONE-D-23-17775R1 

PLOS ONE

Dear Dr. Ravi, 

I'm pleased to inform you that your manuscript has been deemed suitable for publication in PLOS ONE. Congratulations! Your manuscript is now being handed over to our production team.

Kind regards, 

on behalf of

Professor Adekunle Akeem Bakare 

Academic Editor

PLOS ONE